# Sales and Advertising Channels of New Psychoactive Substances (NPS): Internet, Social Networks, and Smartphone Apps

**DOI:** 10.3390/brainsci8070123

**Published:** 2018-06-29

**Authors:** Cristina Miliano, Giulia Margiani, Liana Fattore, Maria Antonietta De Luca

**Affiliations:** 1Department of Biomedical Sciences, University of Cagliari, Cittadella Universitaria di Monserrato-SP 8, Km 0.700, 09042 Monserrato, Cagliari, Italy; cristinamiliano@hotmail.it (C.M.); giulia.margiani35@gmail.com (G.M.); 2CNR Institute of Neuroscience-Cagliari, National Research Council, Cittadella Universitaria di Monserrato-SP 8, Km 0.700, 09042 Monserrato, Cagliari, Italy; lfattore@in.cnr.it

**Keywords:** psychoactive drug marketing, sales channels, Internet, social networks, YouTube, Facebook, Twitter, Instagram

## Abstract

In the last decade, the trend of drug consumption has completely changed, and several new psychoactive substances (NPS) have appeared on the drug market as legal alternatives to common drugs of abuse. Designed to reproduce the effects of illegal substances like cannabis, ecstasy, cocaine, or ketamine, NPS are only in part controlled by UN conventions and represent an emerging threat to global public health. The effects of NPS greatly differ from drug to drug and relatively scarce information is available at present about their pharmacology and potential toxic effects. Yet, compared to more traditional drugs, more dangerous short- and long-term effects have been associated with their use, and hospitalizations and fatal intoxications have also been reported after NPS use. In the era of cyberculture, the Internet acts as an ideal platform to promote and market these compounds, leading to a global phenomenon. Hidden by several aliases, these substances are sold across the web, and information about consumption is shared by online communities through drug fora, YouTube channels, social networks, and smartphone applications (apps). This review intends to provide an overview and analysis of social media that contribute to the popularity of NPS especially among young people. The possibility of using the same channels responsible for their growing diffusion to make users aware of the risks associated with NPS use is proposed.

## 1. Introduction

In the last 10 years, an increasing number of new psychoactive substances (NPS) has flooded the drug market. NPS are drugs of misuse not included in the International United Nations Conventions, which can easily bypass the supply reduction strategies of law enforcement agencies and sanctions related to the use and sale of illicit substances. The advent of NPS has contributed to the appearance and growth of a new “drug scenario” characterized by an increased number of drug users among young people and the consumption of drugs with unknown effects or safety profiles. At the initial stage of the phenomenon, NPS are typically used by a small group of people. After the use of these substances becomes well-known, their widespread marketing through media and Internet sales begins. This sequence of events causes the beginning of an epidemic diffusion that is eventually prevented by law enforcement agencies that perform important actions and fight against the trafficking and sale of NPS. Unfortunately, the subsequent legal control of these substances only initiates the reformulation of NPS, which induces a typical loop that is highly dangerous to public health. In order to understand the full spectrum of the complex issue of NPS, we provide here an updated overview of the specific field of sales and advertising channels of NPS that represent an important ring in this chain of events.

## 2. The Complex Issue of the New Psychoactive Substances (NPS)

NPS are synthetic compounds that are very popular worldwide, as shown by the alarming number of NPS (779) reported between 2008 and 2017 by 111 countries and territories [1,2,3,4]. Designed in order to substitute classical drugs of abuse with legal surrogates, their expansion leads to an endless effort made by governments and law enforcement agencies to try to contain this phenomenon. A mix of features makes them very attractive, including the difficulty to detect them in human fluid samples by standard drug screening test, their ambiguous legal status, and, as in the case of synthetic cannabinoids, the perceived low risk despite their toxic effects and abuse liability [5,6]. Noteworthy, the exponential increase in the market size of these compounds has been facilitated by the World Wide Web (WWW), where information about their purchase and use are shared, advertised, and spread to everyone. In light of the changing scenario for drug marketing and advertising, the aim of this review is to analyze the role of the web in this emerging trend, focusing on social networks and smartphone applications (apps).

In the current world where communication is based on the Internet and social networks, online sites operate on both the surface and deep web [7,8,9,10] to supply NPS labeled as “not for human consumption” and sold as plant fertilizers, incense, bath salts, or with other aliases in order to avoid legislative controls [11]. The dark net plays a key role in this “super safe drug dealing”, which buyers and sellers can access anonymously to provide drugs and pay for them with a virtual wallet [2]. Essentially, a few clicks are enough to supply highly psychoactive substances, cheaply and in a low-risk way [12,13], even through smartphone apps [14,15]. Therefore, NPS can be sold to everyone, including very young people, in complete anonymity and easily avoiding law enforcement [10,16,17]. Along with the emergence of new psychoactive drugs in the world drug market, new concepts are emerging to better describe this new, global phenomenon and its associated health consequences. That is, the term “spiceophrenia” has been proposed by Papanti and collaborators [18] to describe the psychotic symptoms (e.g., hallucinations, delusions) that likely occur in chronic users of synthetic cannabinoids. It has been reported that the use of Spice/K2 drugs may exacerbate psychotic symptoms in vulnerable individuals or trigger psychosis in individuals with no previous history of psychosis [19]. The synthetic cannabinoids present in these products may also induce important adverse neuropsychiatric consequences, including acute and lasting psychosis, since their pro-psychotic effects are likely related to the activity of the CB_1_-receptor on dopaminergic, serotoninergic, and glutamatergic pathways [20]. 

Similarly, within the e-health context, the term “e-addictology” has been recently used to indicate new technologies for assessing pathological dependencies and intervening on addictive behaviors, including computerized adaptive testing, e-health programs, web-based interventions, and digital phenotyping [21]. Importantly, new technologies can profoundly change not only the way illegal drugs are supplied, but they can also improve our understanding of drug addiction and favor the development of new interventions for addictive disorders [21]. 

Because not everyone has the finances or the technical skills to create or manage an Internet site, Facebook is often used as an alternative channel for sales and for advertising the use of these kinds of products [8]. On several drug fora, such as www.drugs-forum.com or www.erowid.org, these compounds are promoted and their subjective effects are discussed [9], but drug-related contents also exist on virtually all social networking sites, picture- and video-sharing services, and drug-dedicated apps. Drug selling through social media has also been reported, often using drug slang and jargon [22]. The changing policy on marijuana use in some states in North America, i.e., legalization of medical cannabis, led to an increased rate of cannabis use both in young people and adults [23,24] even though the causal effect of legalization has not been firmly assessed [25]. On the other hand, for young people, is it difficult to recognize the risk of marijuana consumption if the law allows its use for medical purposes, and this might represents a “gateway of curiosity” [26].

## 3. The Deep Web and the Surface Web: Market Resilience 

By typing a keyword in a search engine query such as Google, Bing, Yahoo, or others, web surfers can obtain a list of results belonging to the “surface web”, while other, nonsearchable contents are referred to as the “deep web” or “invisible web” (see Figure 1). The deep web is often confused with the “dark net”, but the two terms are not synonymous and overlap only partially. Basically, the deep web contains all the information stored online which is not indexed by search engines, with most information hidden simply because it is irrelevant for most users. Access to the deep web does not require special tools and a visitor can use specialized search engines or directories to locate the data for which he or she is looking. The dark web, instead, represents a small part of the deep web containing information hidden on purpose, and it typically requires special tools to enter. Like the surface web, the dark web is scattered among servers around the world and represents the portion of the Internet most frequently known for illicit activities. The most common way to access it is through The Onion Router (TOR) and the Invisible Internet Project (I2P). 

Developed in 2010 by the U.S. Army, the TOR browser is able to encrypt a user’s IP address [27], thus making all the operations untraceable. In this way, the anonymous identities of administrators, sellers, and customers are protected [28,29,30,31] and the safe payment of any illicit good is guaranteed by of the use of cryptocurrencies, mainly bitcoins and litecoins, i.e., virtual money not controlled by government [32]. The peer-to-peer software I2P, instead, was created in 2003 purposely for illegal activities [31] in order to provide anonymous access and to bypass police and law enforcement surveillance. I2P uses a “.i2p” domain, different from TOR’s classic Internet domain (WWW), to allow users to host services by I2P’s homepage. The anonymous status in the web is also maintained by encryption of e-mails, files, and messages using different cryptosystems such as Pretty Good Privacy (PGP), the Amnesiac Incognito Live System, and the Tails [33].

Recently, online drug dealing has started replacing the old way of supplying drugs of abuse. Surfing in both the surface and deep web, it is possible to buy traditional illicit drugs but also temporarily legal NPS [1]. 

In this hidden world, the most famous platform is the Silk Road hub. Born but shutdown by the Federal Bureau of Investigation (FBI) in October 2013, it impressively reappeared after a month under the name Silk Road 2.0 in order to supply to demanding customers [34]. Although Silk Road 2.0 was closed in November 2014, it got back on track in May 2016 and is now available as Silk Road 3.0. Moreover, in recent years, many cryptomarkets became available for buying and selling NPS, including Dream Market [33] and others such as Alphabay, Nucleus, and Valhalla, which were shut down in 2016 and 2017, respectively. In addition, a collection of data from drug fora and blogs on the surface web shows that people who possess the knowledge for using the deep web are also able to access drug marketplaces and buy drugs, including NPS [31]. Since the late 2000s, a number of studies have investigated the online supply of NPS through online shops, among which was the two-year, European Commission-funded “Psychonaut 2002 project”, coordinated by Fabrizio Schifano, that provided a quantitative and qualitative assessment of the online supply of NPS in a time-specific context, i.e.,“snapshot” [35]. More recently, another European project, the I-TREND (Internet Tools for Research in Europe on New Drugs) project, cofinanced by the Drug Prevention and Information Programme of the European Union, monitored the evolution of online shops and online user fora, conducted an online survey focused on NPS users, and, based on the analysis of samples and the exchange of reference standards among laboratories, ultimately produced a “top list” of NPS at the national level [22].

However, since the deep web remains inaccessible to everyone, research of NPS occurs also on the surface web, where several websites are currently selling them, advertising the products as incense, bath salts, fresheners, plant fertilizers, etc. Notably, when inserting into classical searching engines, such as Google, keywords like “legal highs” or “herbal highs”, many websites are listed that offer drugs which are still legal thanks to the time that typically elapses from the appearance of a new substance into the market and its introduction in the list of regulated substances [13]. Few of these websites explicitly sold NPS; gaudy pictures, reduced price for first purchase, proposals for use of new equipment (e.g., vaporizers or smoking pipes), gift ideas, and holidays sales are only some examples of the tricks they use to capture the attention of young consumers. Everyone who is looking for a new sensorial experience and willing to try a psychoactive substance is encouraged to make the purchase with guaranteed secure payment and fast shipment.

## 4. Sharing Information: Drug Fora and YouTube

A large proportion of the world’s population uses social networking websites, especially young adults. Therefore, it is not unexpected that conversations about drug use have transferred onto Internet drug fora and message boards [36]. The nature of conversations on drug fora (e.g., www.drugs-forum.com, www.erowid.org) varies significantly. Indeed, drug fora are used for many purposes, including sharing methods of using drugs and learning about new drugs but also for harm reduction purposes [37]. Notably, many users declare to access drug fora primarily to learn how to handle drugs more safely. As a matter of fact, some users claim to be experts and provide detailed guidelines on doses and routes of administration for each drugs class, advising against dangerous drugs interactions as well [38]. The types of visitors and/or participants to drug fora are diverse, but many of them can be considered recreational users who do not consider themselves drug addicts, do not look for treatment, and are not planning to discontinue drug use [39,40].

Drug fora are also very popular among NPS consumers. They are used to report their experiences of the positive and negative effects of substances and to provide advice on doses, routes of administration, and on how to obtain them easily [39], frequently sharing their favorite substances using pharmacological language.

The impact of conversations on drug fora on drug-use behaviors is not known, but it is reasonable to argue that monitoring such discussions could help policy and law enforcement agencies to identify emerging trends in drug use and markets [41]. In addition to drug fora, it is very common to find trip reports on YouTube, the most popular video-sharing site used by teenagers (among other users), and also on the picture-sharing sites Flickr and Instagram. Previously used to report marijuana-, tobacco-, and alcohol-related experiences [42,43], a number of videos of various NPS are now available on YouTube, in which consumers describe in first person all proven effects including negative aspects of their experiences. Sometimes live streams after ingestion of the drug are posted. Many videos can be classified as “cautionary videos” (better known as vernacular prevention videos), others as “hedonistic/celebratory videos” (but not for crystal meth or heroin), and some are “do-it-yourself” (DIY) videos where, for example, detailed instructions on how to grow your own cannabis are provided [44]. Considering the novelty-seeking propensity of young people, this easily available online information might promote the use of these substances. Concern has been expressed for the potential negative impact of social media content depicting drug use and related behaviors [45].

## 5. Social Networks and Smartphone Apps 

In these times, the way to surf the Internet has radically changed and social networks are the new leaders of this trend, with a large percentage of use by teenagers [17]. According to a recent survey from a leading global information and measurement company [46], Internet users engage longer in social media sites and apps than in activity on any other type of website. The same survey estimated that the social networking giant Facebook has currently more than 1.6 billion registered users, that the video-sharing site YouTube has more than 1 billion active users, while the social streaming site Twitter has more than 500 million registered users worldwide. Given these numbers, all these platforms have inevitably attracted the interest of drug suppliers, which strictly follow their evolution and diffusion among young people over time. In the last few years, several social networks have acquired important roles in market places for both NPS and illicit drugs [33] (see Figure 2). Simply looking on Facebook, it is now possible to find information and direct links to proceed with the purchase of several NPS or to simply share your experiences in groups created for drug-users only. Even the picture- and video-sharing service Instagram, despite its different use compared to the more famous Facebook, is used to look for new possible customers [47]. Several ambiguous profiles are used to post pictures of their products with hashtags such as #cannabiseeds, #headshop, #herbalicense, and #over18sonly. In 2014, drugabuse.com, for example, published on Instagram an infographic documenting drug-dealer activity [48].

On Twitter, by simply typing #legalhighs, it is possible to buy “the blue stuff”, otherwise known as methamphetamine, of the famous *Breaking Bad* television series as well as other #ResearchChemicals, with free shipping offers and credit/debit card payments accepted. All the users, indeed, can be part of a #highsociety, allowing them to share their #proudstoners daily states of mind. Very recently, epidemiologists and linguistic scientists used Twitter to test the feasibility of producing a fully-automated “drug term discovery” system capable of tracking emerging NPS terms in real time [49], which confirms that data collected on Twitter may be used to explore trends in NPS selling and use [50]. Along with other cyber drug communities (e.g., blogs, drug fora, Facebook), Twitter allowed the identification and characterization of a new generation of NPS users, the so-called “e-psychonauts”, that considered themselves as psychedelic researchers, mind navigators, or chemicals experimenters [51].

In a technological world where about 2.4 billion people use a smartphone and an increasing number of people use apps, drug dealers adapt their activity accordingly and create simple apps that make buying any kind of psychoactive substance easier, including NPS. Although some apps are designed to prevent drug use, such as “Your Face on Meth” (where you can upload your picture and see its physical degradation over time potentially resulting from using methamphetamine), many apps are created specifically to promote drug use [52].

In North America, the number of cannabis-related smartphone apps is very impressive. In 2014, the number of apps returned from searches using terms like “cannabis” and “marijuana” were 124 and 218, respectively, in Apple’s Store, and 250 for both on Google Play [14]. These apps have several content codes, contain information on different cannabis strains and synthetic cannabinoids mixture (e.g., K2, Spice), advice for growing cannabis, and recipes for cooking “special meals”. Several apps create a connection with medical marijuana doctors to obtain a prescription, while other apps, such as Eaze, Nugg, Meadow, and Weed Maps, offer to trace medical dispensaries of marijuana, indicating to users the closest spot based on their location in addition to finding the closest doctor that will recommend medical marijuana [14,15]. Additionally, using the app High There, for instance, it is possible to match people to smoke together, while other apps, similar to Instagram and used mostly in Europe and the United States, are utilized for posting photos, videos, or texts related to marijuana or psychoactive substances. Noteworthy, apps useful for making untraceable calls to drug dealers are becoming very popular. 

## 6. Conclusions

New psychoactive substances are very popular among young people and online communities, but very little information about toxicological and side effects are available on the Internet. The web-based open sale of unregulated NPS has shown a steady increase in recent years; the easy availability of NPS and the fluctuating dynamics of this new drug market represent a public health concern and an intricate regulatory issue. Research in the field is increasing, and several groups of clinical and preclinical researchers worldwide are investigating the central and peripheral effects of synthetic cannabinoids and synthetic cathinones [1,53,54,55,56,57,58], synthetic opioids and ketamine-like compounds [59,60], and many others [61,62]. Yet, it is fundamental to share scientific evidence on risks related to the consumption of these compounds using the same channels that promote them. Analyses of social media may represent a new approach to uncover and track changes in drug terms and markets in near real time. In conclusion, the NPS phenomenon is intricate and still very difficult to control. Using the same channels responsible for their growing diffusion to disseminate information and scientific knowledge about the risks associated with their use could represent a potential new approach to limit the diffusion of these dangerous substances.

## Figures and Tables

**Figure 1 brainsci-08-00123-f001:**
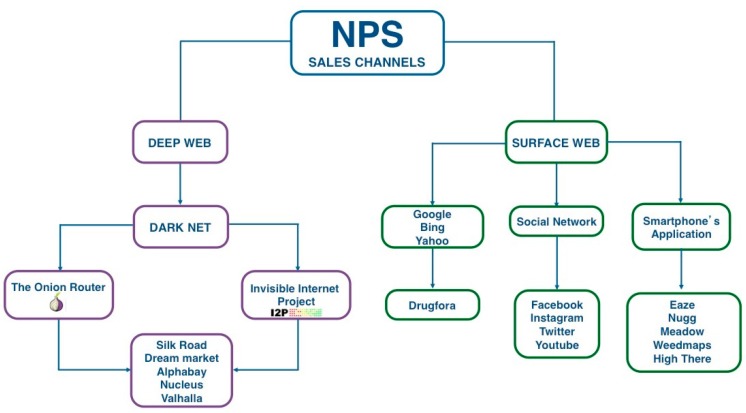
NPS marketing, advertising, and communication network.

**Figure 2 brainsci-08-00123-f002:**
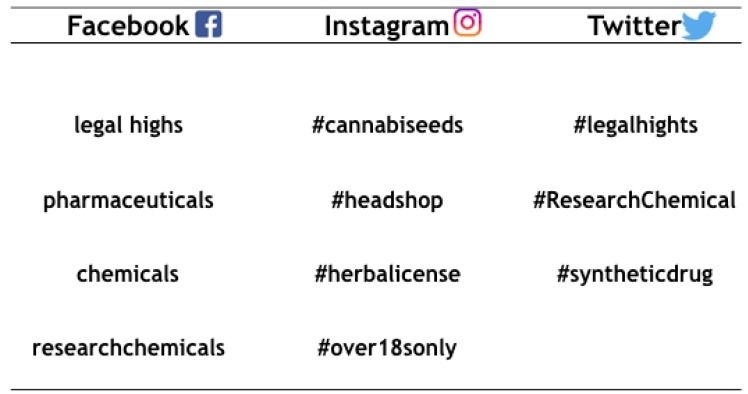
Keywords and hashtags in social networks with explicit content on NPS.

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
