# Peer review of "Sales and Advertising Channels of New Psychoactive Substances (NPS): Internet, Social Networks, and Smartphone Apps"

_brainsci, 2018, doi:10.3390/brainsci8070123_

Round 1
Reviewer 1 Report
This is an interesting review that adds some value to previous reviews that only analyzed the surface web.
Author Response
We thank the reviewer for his/her constructive comments and effort.
Reviewer 2 Report
Review: Sales and advertising channels of New Psychoactive Substances (NPS): Internet, social networks and smartphones’ apps
General comments:
The manuscript provides a brief overview of the current online channels that have contributed to the dramatic evolution of the drug scene.
The abstract and conclusions are too broad and vague. They are not capturing or summarising the important points discussed in this manuscript. Some up-to-date references are missing from this manuscript. Grammar and punctuation need to be reviewed. For example: abstract Page 1 – 3rd line: where should be replaced by which; abstract page 1 – 7th line: intents should be replaced by intends; key words: should be either psychoactive drugs’ marketing or psychoactive drug marketing, etc. Also the use of very long sentences e.g. introduction, second paragraph, first 4 lines.
Abstract:
Too broad and vague, not capturing the main findings.
Introduction:
I suggest you add a statement to set the scene and explain what will be discussed.
Reference 2 and the numbers if the first three lines are out of date.
Page 1 Line 12: please add “a” before “few”
Deep Web and Surface Web: The Market Resilience
“Basically, the deep web contains all of the information stored online but not indexed by search engines, with most of information being hidden simply because irrelevant for most users.” Please review grammar.
“Access to the deep web does not require special tools and one visitor can use specialized search engines or directories to locate the data that is looking for.” Please review grammar.
“Like the surface web, also the dark web is scattered among servers around the world, represents the portion of the Internet most frequently known for illicit activities and the most common way to access it is through The Onion Router (TOR) and the Invisible Internet Project (I2P).” Please review grammar.
Sharing the information: DRUGFORA and YOUTUBE
“which is the best route of administration for a specific drug, the safer drug doses to use, which drug-drug interactions avoid because too dangerous, etcetera [36].” Please review grammar.
“Drug fora are very popular also among consumers of NPS, which usually use them to report their experiences on positive and negative effects of substances, provide advices on doses, routes of administration and how to obtain them easily [7], frequently sharing their favorite substance and using a pharmacological language.” Avery long sentence. Also, please review grammar.
Please add a caption for Figure 1. Please be consistent Fig. or Figure.
I suggest that Figure 1 should be in the section titled: Deep Web and Surface Web: The Market Resilience
“Noteworthy, some apps like are becoming very popular to make untraceable calls to contact drug dealers”. I think there is a missing word in this sentence.
Conclusions:
Page 5 First line: “famous”? do you mean popular?
Please be specific and extract your main conclusions.
References:
Please edit as appropriate (e.g. see ref 54, 55)
Please consider reviewing more up-to-date published papers
Author Response
We thank the Reviewer for his/her constructive comments and helpful suggestions. We made our best to address the requests. Please, find changes highlighted in yellow in the text.
COMMENT: Too broad and vague, not capturing the main findings.
REPLY: As suggested, the abstract has been modified in order to be more clear and explicit.
COMMENT: Introduction: I suggest you add a statement to set the scene and explain what will be discussed.
REPLY: A new introduction of 14 lines has been added and is followed by a new paragraph titled “The complex issue of the Novel Psychoactive Substances (NPS)” where minor changes have been also made.
COMMENT: Reference 2 and the numbers if the first three lines are out of date.
REPLY: Done.
COMMENT: Page 1 Line 12: please add “a” before “few”.
REPLY: Done.
COMMENT: Deep Web and Surface Web: The Market Resilience
“Basically, the deep web contains all of the information stored online but not indexed by search engines, with most of information being hidden simply because irrelevant for most users.” Please review grammar.
“Access to the deep web does not require special tools and one visitor can use specialized search engines or directories to locate the data that is looking for.” Please review grammar
Like the surface web, also the dark web is scattered among servers around the world, represents the portion of the Internet most frequently known for illicit activities and the most common way to access it is through The Onion Router (TOR) and the Invisible Internet Project (I2P).” Please review grammar.
REPLY: Title and contents have been check for grammar by a native English-speaker.
COMMENT: Sharing the information: DRUGFORA and YOUTUBE
“Which is the best route of administration for a specific drug, the safer drug doses to use, which drug-drug interactions avoid because too dangerous, etcetera [36].” Please review grammar.
“Drug fora are very popular also among consumers of NPS, which usually use them to report their experiences on positive and negative effects of substances, provide advices on doses, routes of administration and how to obtain them easily [7], frequently sharing their favorite substance and using a pharmacological language”. A very long sentence. Also, please review grammar.
REPLY: Title and contents have been check for grammar by a native English-speaker.
COMMENT: Please add a caption for Figure 1. Please be consistent Fig. or Figure.
REPLY: As requested, a caption for figure 1 has been added. The abbreviation Fig. has been chosen.
COMMENT: I suggest that Figure 1 should be in the section titled: Deep Web and Surface Web: The Market Resilience.
REPLY: Done.
COMMENT: Noteworthy, some apps like are becoming very popular to make untraceable calls to contact drug dealers”. I think there is a missing word in this sentence.
REPLY: The sentence has been modified as follow: “Noteworthy, apps useful to make untraceable calls to drug dealers are becoming very popular.”
COMMENT:
Conclusions: Page 5 First line: “famous”? do you mean popular? Please be specific and extract your main conclusions.
REPLY: Famous has been replaced by popular. New main conclusions have been added.
COMMENT:
References: Please edit as appropriate (e.g. see ref 54, 55). Please consider reviewing more up-to-date published paper.
REPLY: Done.